# Subependymal Giant Cell Astrocytomas in Tuberous Sclerosis Complex—Current Views on Their Pathogenesis and Management

**DOI:** 10.3390/jcm12030956

**Published:** 2023-01-26

**Authors:** Chao Gao, Bernadeta Zabielska, Fuyong Jiao, Daoqi Mei, Xiaona Wang, Katarzyna Kotulska, Sergiusz Jozwiak

**Affiliations:** 1Department of Rehabilitation Medicine, Henan Children’s Hospital, Children’s Hospital Affiliated to Zhengzhou University, Zhengzhou 450018, China; 2Faculty of Medicine, Medical University of Warsaw, 02-091 Warsaw, Poland; 3Pediatric Neurology Section, Children’s Hospital of Shaanxi Provincial People’s Hospital, Xi’an 710068, China; 4Department of Neurology, Henan Children’s Hospital, Children’s Hospital Affiliated to Zhengzhou University, Zhengzhou 450018, China; 5Henan Key Laboratory of Children’s Genetics and Metabolic Diseases, Henan Children’s Hospital, Children’s Hospital Affiliated to Zhengzhou University, Zhengzhou 450018, China; 6Department of Neurology and Epileptology, The Children’s Memorial Health Institute, 04-730 Warsaw, Poland; 7Research Department, The Children’s Memorial Health Institute, 04-730 Warsaw, Poland

**Keywords:** subependymal giant cell astrocytoma (SEGA), tuberous sclerosis complex (TSC), mTOR inhibitors, management, recommendations

## Abstract

Introduction, Tuberous sclerosis complex (TSC) is an autosomal-dominant disorder caused by mutations inactivating TSC1 or TSC2 genes and characterized by the presence of tumors involving many organs, including the brain, heart, kidneys, and skin. Subependymal giant cell astrocytoma (SEGA) is a slow-growing brain tumor almost exclusively associated with TSC. State of the Art: Despite the fact that SEGAs are benign, they require well-considered decisions regarding the timing and modality of pharmacological or surgical treatment. In TSC children and adolescents, SEGA is the major cause of mortality and morbidity. Clinical Implications: Until recently, surgical resection has been the standard therapy for SEGAs but the discovery of the role of the mTOR pathway and the introduction of mTOR inhibitors to clinical practice changed the therapeutic landscape of these tumors. In the current paper, we discuss the pros and cons of mTOR inhibitors and surgical approaches in SEGA treatment. Future Directions: In 2021, the International Tuberous Sclerosis Complex Consensus Group proposed a new integrative strategy for SEGA management. In the following review, we discuss the proposed recommendations and report the results of the literature search for the latest treatment directions.

## 1. Introduction

Tuberous sclerosis complex (TSC), also known as Bourneville–Pringle syndrome, is an autosomal dominant, potentially devastating, a neurocutaneous disorder characterized by the development of hamartomas in several organs, including the brain, kidneys, lungs, heart, eyes, and skin [1]. With an incidence of about 1:6000 in the general population TSC is the second most common neurocutaneous disorder [2].

TSC manifestations are caused by mutations in either *TSC1* (on chromosome 9q34) or the *TSC2* gene (on chromosome 16p13.3), both recognized as tumor suppressors. Loss of *TSC1* or *TSC2* gene function results in the overactivation of the mTOR (mammalian target of rapamycin) signaling pathway leading to impaired control of cell growth, differentiation, and proliferation [3]. In general, patients with the *TSC2* mutation are characterized by a more severe course of the disease and earlier development of symptoms than individuals with *TSC1* mutations [4]. There are many scientific studies based on animal studies showing that the mentioned mutation is also a more common cause of severe phenotypic features, including the intensity and frequency of epilepsy and other neurological symptoms. On a similar note, patients carrying a PKD mutation with an accompanying TSC2 mutation are at risk for an earlier onset of polycystic kidney disease and its more acute course.

The most common neurological manifestation associated with this condition is epilepsy, which appears in about 90% of patients. Early onset of epilepsy in the first year of life is frequently accompanied by TSC-associated neuropsychiatric disorders (TAND), especially intellectual disability and autistic behavior [5,6].

*TSC2* gene mutations are four times more common than *TSC1* gene mutations in the TSC population; however, in familial cases which comprise about one-third of all TSC individuals, the proportion is 1:1 [7].

TSC-related neoplasms are generally benign, non-infiltrative and classified by the World Health Organization as grade I, and the risk of mortality and morbidity is associated with their volume and location [8]. One of the characteristics of TSC is the development of age-specific tumors in various organs over time, starting from the prenatal period. Cardiac tumors and cortical brain tubers develop prenatally, skin lesions typically start to grow in toddlers, brain and kidney tumors in older children and adolescents, and lung involvement is present usually not earlier than in adults [9]. Interestingly, some of the tumors typically do not grow beyond a specific age: for example, cardiac tumors usually tend to regress in infants and brain tumors do not increase their size in adults. Other lesions, such as kidney and liver AMLs, skin manifestations, or pulmonary lymphangioleiomyomatosis (LAM) are characterized by a constant tendency to grow. Despite the benign nature of the tumors, TSC is associated with increased mortality in children and young adults [10,11]. Analysis of the Tuberous Sclerosis Alliance TSC Natural History Database of 2233 TSC patients revealed that the median age of death of 31 decedents was 28 years [10].

The diagnosis of TSC is established on the basis of clinical manifestations or pathological genetic findings. The newest diagnostic criteria of TSC have been recently published (Figure 1) [12]. In 2021, the International TSC Clinical Consensus Group reaffirmed the importance of independent genetic diagnosis. Identification of a pathogenic variant in *TSC1* or *TSC2* is sufficient for the diagnosis of TSC regardless of clinical findings. The diagnosis may be also established if two major or one major and two minor clinical criteria are present. Currently, more and more patients are diagnosed prenatally due to cardiac and brain lesions which may be revealed in routine fetal ultrasonography and magnetic resonance imaging (MRI) [13]. Disclosure of a cardiac tumor and a brain lesion specific for TSC is consistent with a prenatal diagnosis of TSC.

In this review, we summarize current opinion on the pathology, development, and management of subependymal giant cell astrocytoma (SEGA). These brain tumors are characteristic of TSC and present a major cause of morbidity and mortality in TSC patients.

## 2. Epidemiology of SEGA

Usually, the diagnosis of SEGA is established during MRI surveillance or, less frequently nowadays, due to the symptoms of related acute hydrocephalus in a TSC patient. Incidentally, a histopathological examination may demonstrate SEGA characteristics in a patient without a TSC diagnosis. However, in some of these cases, further extended clinical examination may reveal other signs of the disease and confirm the diagnosis of TSC [14].

SEGA is a tumor representing 1% to 2% of all pediatric brain tumors and is usually associated with TSC [15]. In the existing literature, the prevalence of SEGA in TSC varies from 6 to 25% [16].

In a large cohort of 2223 children and adult patients registered in the TOSCA database, 510 (24.4%) patients reported having SEGA [17]. A similar prevalence of 20% has been established by Adriaensen et al. [18]. In this retrospective cross-sectional study based on brain CT analysis, the tumor size ranged from 4 to 29 mm and in 9 of the 43 cases (21%) the lesions were bilateral. Interestingly, a higher incidence of SEGA was found in men (22%) than in women (18%) [18].

SEGAs are more frequently reported in children than in adults. An online survey conducted among 676 TSC patients and caregivers showed that these tumors were present in 26.3% of patients aged 18 years or younger and in 13.9% of older patients [19]. This effect might result from effective SEGA treatment in children and adolescents and, on the other hand, from the increased mortality in patients with SEGA. There are very few reports on SEGAs growing after 25 years of age, thus brain MRI surveillance is recommended only in younger patients [12].

Typically, SEGA is diagnosed in patients between 5 and 15 years of age [20], but earlier development of the tumor, even prenatally, has also been reported [21,22]. Children with very early presentation of SEGA are also at high risk of severe epilepsy, intellectual disability, and autistic spectrum disorder [23]. There is a strong correlation between early SEGA development and the *TSC2* genotype. Congenital SEGA is almost exclusively associated with the *TSC2* gene mutation. All 10 patients with congenital SEGA reported by Chan et al. (2021) and all eight infants with genetically confirmed TSC in the study of Kotulska had *TSC2* mutations [23,24].

Patients with large *TSC2* gene deletions affecting the adjacent *PKD1* gene were reported to be at the highest risk of early SEGA development. Such large deletions account for 2–3% of all TSC cases but were found remarkably more frequently in congenital SEGA cases—in 37.5% (3 out of 8 patients) in Kotulska et al.’s group [24] and in 10% (1 out of 10) in Chan et al.’s study [23].

## 3. Pathology and Pathogenesis of SEGA

Cortical tubers, subependymal nodules (SENs) and SEGAs represent the main neuropathological lesions in patients with TSC. Contrary to non-growing cortical tubers, SENs and SEGAs may increase their size over time and lead to life-threatening manifestations [22].

SENs are considered the precursor lesions of SEGAs. They are usually small lesions seen in over 90% of TSC patients in subependymal zones of lateral ventricles of the brain. It has been documented in multiple studies that non-calcified SENs may slowly transform into SEGAs [20,25,26,27]. However, the exact moment of this transition is not fully defined.

Despite several workshops and meetings of TSC experts in the USA and Europe, it was not possible to obtain a full consensus regarding the definition of SEGA. Histopathologically SENs and SEGAs are indistinguishable (Figure 2) [28]. They are composed of large ganglion-like astrocytes, with mixed glio-neuronal features and elevated levels of cytoplasmic phospho-S6K, phospho-S6, and phospho-Stat3 proteins downstream of mTORC1 [29]. Both SENs and SEGAs share unique expression profiles and their characteristic molecular signature includes dysregulation of inflammation and the extracellular matrix [30,31]. Stable SENs and tubers transforming into SEGA are also not clearly differentiable in MR images; however, rapid growth or gadolinium enhancement could be suggestive of SEGA’s transformation [27,32]. Most authors agree that the documented growth of lesions over 1 cm, even in the absence of enhancement on MRI, is consistent with SEGA diagnosis [33]. There is no consensus on the time in which its growth should be documented. For example, it is not known if SEGA should be diagnosed in a patient born with a 1-cm lesion when the tumor does not show any increase in size for several years.

The pathogenesis of the transformation of SEN into SEGA is not yet fully understood. The concept that SEGAs develop from SEN is widely accepted, but the mechanisms underlying their progressive growth are unknown so far. One of the hypotheses says that it might be related to the location of SENs, as SEGAs most typically arise at the caudothalamic groove adjacent to the foramen of Monro. The increased flow of nutritional factors through the foramen of Monro may accelerate the growth of SEN and its transformation into SEGA. However, SEGA may also grow in other locations, especially in the third ventricle. The molecular background of SEGAs growth is also uncertain. Several studies evidenced a second-hit inactivation of *TSC1* or *TSC2* in SEGAs, suggesting that one contributor to SEGA development is the complete loss of a functional tuberin-hamartin complex and the subsequent mTORC1 activation. This is in line with the Knudson two-hit mechanism of tumor development and was also found in other lesions related to TSC. In the study of Bongaarts [29] nineteen of 34 (56%) brain tumor samples had mutations in *TSC2*, 10 (29%) had mutations in *TSC1*, while five (15%) had no mutation identified in *TSC1/TSC2.* The majority of these samples had a loss of heterozygosity in the same gene in which the primary mutation was identified. Bongaarts et al. [34] also studied the role of BRAF kinase as an alternative factor contributing to SEGA development. They assessed the prevalence of the *BRAF* V600E mutation in a large cohort of 58 TSC-related SEGAs and found no evidence of either *BRAF* V600E or other mutations in *BRAF*.

Currently, there is an increasing body of evidence that the transformation of SENs to SEGAs may be additionally triggered by the upregulation of genes related to the MAPK pathway. Siedlecka et al. [28] studied the expression of Akt, Erk, and mTOR pathways in SEN and SEGAs. SEN and one SEGA sample were obtained from the same patient. They found significant upregulation of p-Erk, p-Mek or p-RSK1 in SEGAs, but not in the SEN sample. At the same time p-Akt, p-GSK3β and p-PDK1 were upregulated in both SEN and SEGA from the same TS patient. The authors proposed the following sequence of events: the activation of PI3K/Akt leads to the upregulation of mTOR. These two events result in SEN formation but are not enough for the transformation of the cell into SEGA. According to this hypothesis, it is the Erk activation that triggers the growth of SEN into SEGA [28] (Figure 3).

Bongaarts et al. [34] also identified an upregulation of genes related to the MAPK pathway in SAG samples. ERK activation was confirmed on the protein level and both the ERK inhibitor U0126 and rapamycin were able to decrease the proliferation of SEGA cells in vitro. Additionally, genes related to the Ragulator complex, a complex activating both the MAPK/ERK and mTORC1 pathway, were overexpressed in SEGA compared to control tissue.

In summary, it is likely that SEGAs growth might be caused not by one but several triggering factors. Restricted numbers of available SENs specimens are an important limitation of further studies.

## 4. Non-TSC Associated SEGA

In recent years there has been an increasing number of reports of SEGA diagnosis in “healthy” patients. The occurrence of SEGA in non-TSC patients is very rare, and those patients should undergo a detailed clinical workout for other features of TSC including *TSC1/TSC2* genetic analysis in the search for “forme fruste” of TSC. The patients with TSC manifestations not sufficient for definite diagnosis of the disease may have low-level somatic mosaicism for *TSC1/TSC2* mutations detectable only with deep sequencing methods [35].

However, SEGAs were reported also in otherwise healthy people, in whom all examinations for TSC, including genetic testing of DNA of epithelium of buccal mucosa, urine sediment, and blood cells, are negative. Ichikawa et al. [36], for example, described a 20-year-old woman with clinical manifestations of a brain tumor. She underwent brain surgery and a histological examination of the tumor was consistent with SEGA diagnosis. Molecular analysis of the tumor confirmed loss of heterozygosity and allelic mutation of *TSC2* gene. The clinical workout for TSC manifestations was negative. Her DNA analysis from peripheral blood, buccal mucosa, urinary sediment, nail and hair revealed no *TSC1/TSC2* pathogenic mutation. From a genetic point of view, these isolated SEGAs are thought to result from two purely somatic mutations in one of the TSC genes (*TSC1* or *TSC2*) limited to the tumor. Similar mechanisms were reported in sporadic retinoblastoma and several other cancers. It must be emphasized that in such cases it is crucial to rule out low-level mosaicism in other tissues since its presence implies specific follow-up and a possible risk of transmission to offspring.

## 5. Clinical Presentation of SEGA and Patients’ Surveillance

Usually, SEGAs grow slowly and do not produce clinical symptoms for a long time, but on longer follow-up, by the occlusion of the foramen of Monro, they may lead to hydrocephalus and symptoms of increased intracranial pressure (Figure 4) [16,37,38]. These symptoms include headaches, nausea, vomiting, especially in the morning or during the night, blurred vision, changes in behavior, and new or worsened seizures. In patients with disabilities, early signs of increased intracranial pressure may be easily overlooked, so monitoring of SEGA with neuroimaging is now recommended.

In the past, SEGAs were frequently diagnosed in patients presenting with symptoms of increased intracranial pressure. Nowadays, due to the implementation of MRI surveillance, the majority of tumors are diagnosed at an early stage allowing effective treatment [12,32]. In about 15–40% of patients, SEGAs may develop bilaterally or at several different locations [20].

In 2021, the International Tuberous Sclerosis Complex Consensus Group updated the diagnostic criteria and treatment recommendations [12]. Every TSC patient below the age of 25 years should undergo regular brain MRI examinations every 1–3 years to assess the presence of SEGA and evaluate a change in the size of the tumor. Patients with large or growing SEGA, or with SEGA causing ventricular enlargement who are asymptomatic, should undergo MRI scans more frequently and the patients and their families should be educated regarding the potential new symptoms [12]. Because the growth of tumors seems to slow down dramatically after 25 years of age, frequent MRIs are not recommended routinely but should be performed according to individual needs. However, in the adult TSC population included in the TOSCA study, the continued growth of SEGA was reported in 21% of patients, predominantly with the *TSC2* genotype and 2.4% were newly diagnosed during adulthood, the oldest of whom was 57 years old. Most of the patients had mutations in the *TSC2* gene [39]. Therefore, patients with SEGA diagnosed in childhood should undergo regular brain imaging also beyond the age of 25 years.

Today, there is a great variety of methods of tumor size assessment in MRI studies. The most popular, traditional manner is the planimetric methodology of volumetric analysis but there are also semi-automatic ways of tumor scanning. ITK-Snap (pixel clustering, geodesic active contours, region competition methods), 3D Slicer (level-set thresholding), and NIRFast (k-means clustering, Markov random fields) have proved to facilitate the appropriate assessment of SEGA growth [40].

## 6. Surgical and Pharmacological Treatment of SEGAs

Currently, two treatment modalities are available for growing SEGAs: surgical or pharmacological interventions with mTOR inhibitors (mTORi). Before the era of mTORis, neurosurgical resection of SEGAs was the standard therapy in patients with TSC [32]. Nowadays, according to recommendations of the International Tuberous Sclerosis Complex Consensus Conference surgical resection should be performed for acutely symptomatic SEGA (sometimes with cerebral spinal fluid shunt insertion). For growing but otherwise asymptomatic SEGA either surgical resection or medical treatment with mTORi may be used.

The most common resection routes are transfrontal transcortical and interhemispheric transcallosal [41]. Early and total removal of the tumor is associated with a better prognosis. The experience of the surgical team is also an important factor in favorable outcomes.

Postoperative morbidity and mortality increase when SEGA invades the neighboring structures as well as in bilateral and larger tumors. In a review of 263 TSC patients affected by SEGA gross total resection was achieved in 81.1% of cases, and mortality and permanent morbidity were 4.9% and 6.1%, respectively. A cerebrospinal fluid shunt was needed in 81 patients (30.8%). Tumors regrew in 11.5% of cases, and in most cases, the regrowth was seen when partial tumor resection was performed [42]. Subtotal resection results in a very high probability of regrowth, and medical treatment should be preferred in cases when total resection is not feasible or the size of tumors exceeds 4 cm [20,43]. In the largest series of 64 resected SEGAs surgical treatment of tumors >4 cm or bilateral tumors was associated with a very high risk of complications of 73% and 67%, respectively [20]. Surgery-related complications (hemiparesis, hydrocephalus, memory deficits and intracranial bleeding) were most commonly reported in children younger than 3 years of age (Table 1) [42].

Minimally invasive surgical techniques may increase surgical safety in selected patients.

Endoscopic tumor removal has been more extensively considered in recent years; however, its main limitations are tumor size (<3 cm) and broad attachment of the tumor to the basal ganglia. The advantages of endoscopic management are also the possibility to add septostomy to tumor resection [44]. Among the less invasive surgical techniques, the keyhole concept method is also worth highlighting. It involves accessing deep intracranial lesions through the minimum craniotomy, which significantly reduces the operational risk and improves the cosmetic effect.

Laser interstitial thermal therapy (LITT) is the more recently considered option. It has similar limitations to the endoscopic approach (tumor size < 2 cm, broad attachment of the tumor to the basal); moreover, there is a risk of acute hydrocephalus and edema of basal ganglia, so active hydrocephalus is a contraindication to LITT in SEGAs [44]. This new minimally invasive technique is very promising; however, so far there are no data on the long-term results of LITT in SEGA. Given that radiation of tumors associated with Gamma Knife Stereotactic Radiosurgery (GK-SRS) may promote malignant malformation, the use of this therapy in the treatment of SEGA is very limited [45,46].

SEGAs were the very first manifestations of TSC, in which the mTORi have been used. Currently, two mTORi, everolimus and sirolimus, are widely used in the treatment of SEGAs. Several prospective trials documented successful SEGA regression with both agents, but only everolimus was used in a controlled, randomized study [47,48,49]. In the double-blind EXIST-1 study, everolimus significantly decreased the volume of SEGAs by at least 50% in 35% of patients after 6 months of treatment [49]. Longer follow-up of these patients) resulted in even higher numbers—62% of patients achieved at least a 50% reduction in the tumor volume after 4 years of treatment [50]. Everolimus is currently approved by FDA and EMA for the medical treatment of SAGAs in patients not eligible for surgical treatment.

Both mTOR inhibitors, everolimus and sirolimus, share the same main mechanism of reducing the activity of the mTOR pathway. However, due to their different clinical profile, patients may tolerate one drug better than the other and may have a greater response and/or fewer side effects with everolimus versus rapamycin or vice versa [51].

International Tuberous Sclerosis Complex Consensus Conference widely recommended the use of mTOR inhibitors for patients with asymptomatic and growing SEGA, as well as in patients with mild and moderate symptoms who are not eligible for surgery or prefer medical treatment. There are also several additional situations when the use of mTOR inhibitors may be particularly useful. There is an increasing number of reports on the presurgical administration of mTOR inhibitors in large SEGAs or in tumors invading deep brain structures such as the hypothalamus or thalamus. The neoadjuvant therapy may enable tumor size reduction and gross total resection. Moreover, intraoperative examination in such cases showed fewer features of hemorrhage and clearly differentiated borders of the tumor [52]. The use of mTOR inhibitors can also be found in the management of microscopic residual tumors. Such treatment is used as a neoadjuvant therapy in cases of poorly operated tumors or in the absence of absolute certainty as to their annihilation. Such a maneuver significantly reduces the patient’s risk of reoperation and the associated inconvenience.

Contrary to general indications, mTORi has been also reported to be used in acute obstructive hydrocephalus, including five patients who had clinical signs of increased intracranial pressure. Such an approach allowed significant tumor shrinkage and ventriculomegaly diminution without the use of a CSF shunt [53]. Other studies included patients with moderate hydrocephalus who were not eligible for surgery. Treatment with everolimus resulted in an improvement in ventricular dilatation [54,55].

Finally, mTORi was used for the prevention of tumor recurrence after subtotal resection of SEGA. Franz et al. reported persistent volume reduction in 3 out of 4 patients during long-term pharmacological treatment [56].

During the decision-making process on SEGA treatment, one should also consider the impact of mTOR inhibitors on other manifestations of TSC. Application of mTORi may also decrease seizure frequency in patients with refractory focal seizures, cause regression of cardiac rhabdomyomas in infants, as well as reduce the size of kidneys angiomyolipomas and improve facial angiofibromas [49,57]. Treatment with mTOR inhibitors can additionally improve autistic behavior. Yui et al. reported marked improvement in social interaction and verbal and non-verbal communication in four young TSC patients treated with an mTOR inhibitor [58]. A concomitant increase in ceruloplasmin and transferrin was observed.

Despite these beneficial effects, treatment with mTORi is associated with several adverse events.

First, the therapy should be continued in the long term. Prospective clinical trials showed that the discontinuation of treatment leads to incomplete response and regrowth of the tumor [59]. Even if the growth potential of the tumor seems to be weaker after 25 years of age, there are some reports of SEGA diagnosis after this age; therefore, long-term awareness is necessary. So far, there are no reports on the successful discontinuation of mTOR therapy in patients older than 25 years of age with no regrowth of the tumor.

Second, side effects of mTORi are seen in up to 96% of patients [49]. The most frequent adverse events are usually mild and include aphthous ulcers, stomatitis, pyrexia, nasopharyngitis, fatigue, otitis media and upper respiratory tract infections [49]. There is also an increased risk of reduction in insulin secretion and insulin resistance [60,61]. Some patients may develop severe adverse events from mTOR inhibitors, including pneumonitis, sepsis, and amenorrhoea [59].

Third, it should be remembered that some medications, increasing the activity of cytochrome P450 may interfere with mTORi treatment. Such medicines as cannabidiol, rifampicin, some antibiotics and antiepileptic drugs may change the therapeutic levels of everolimus and sirolimus, which is why many laboratory tests are required to initiate and continue the treatment with mTOR inhibitors in patients with TSC.

Adverse events reported in the first years of the treatment become less pronounced with increasing periods of treatment [50]. However, the very long-term side effects of mTORi are not known. The pros and cons of medical and surgical treatment of SEGA are presented in Table 2. The proposed diagnostic algorithm is also demonstrated in Figure 5.

There are very few studies describing protocols for long-term treatment. As the therapy is dose-dependent, the dosage of the drug may be reduced in order not to expose the patient to side effects. The EMINENTS Study demonstrated that in people treated with everolimus effectively with the full dose, the dose of the drugs may be reduced with the sustained therapeutic effect [62].

In the era of new and emerging therapeutic options, the decision on the treatment of SEGA may be more complex and should be taken by a multidisciplinary team, including neurologists and neurosurgeons together with the patients and their caregivers. The decision should take into account all patient’s needs, the expected duration of the therapy and the experience of the surgical team.

## 7. Conclusions

Today, thanks to recommended MRI follow-up, SEGA is diagnosed before the onset of neurological symptoms in the majority of TSC patients enabling early treatment. However, the optimal timing of SEGA treatment is not defined and requires further studies.Surgery and medical treatment with mTORi are used to treat SEGA and there is no consensus on the priority of one treatment over another. Currently, everolimus is approved for the treatment of unresectable tumors, but increasing evidence shows that it can be successfully used as a neoadjuvant therapy in combination with surgery.The treatment decision should be taken during the multidisciplinary discussion concerning the individual needs and challenges of the patient.

## Figures and Tables

**Figure 1 jcm-12-00956-f001:**
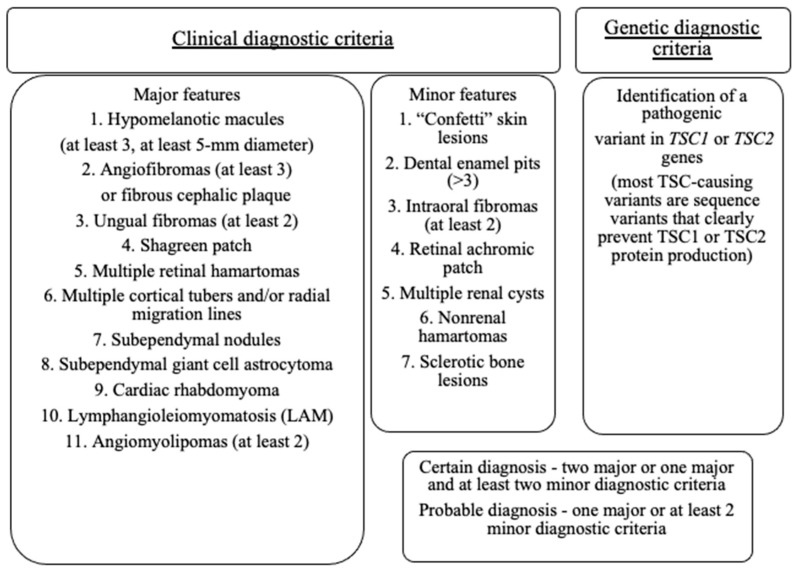
Updated Diagnostic Criteria of Tuberous Sclerosis Complex According to Northrup et al. (2021) [12].

**Figure 2 jcm-12-00956-f002:**
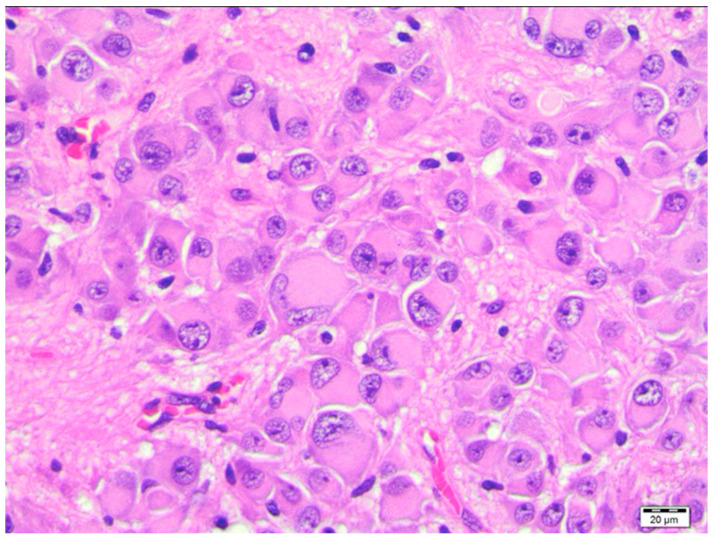
SEGA. Large cells resembling ganglion cells and gemistocytic astrocytes. HE staining. (Courtesy of prof. Wiesława Grajkowska, Pathology Department, The Children’s Memorial Health Institute, Warsaw).

**Figure 3 jcm-12-00956-f003:**
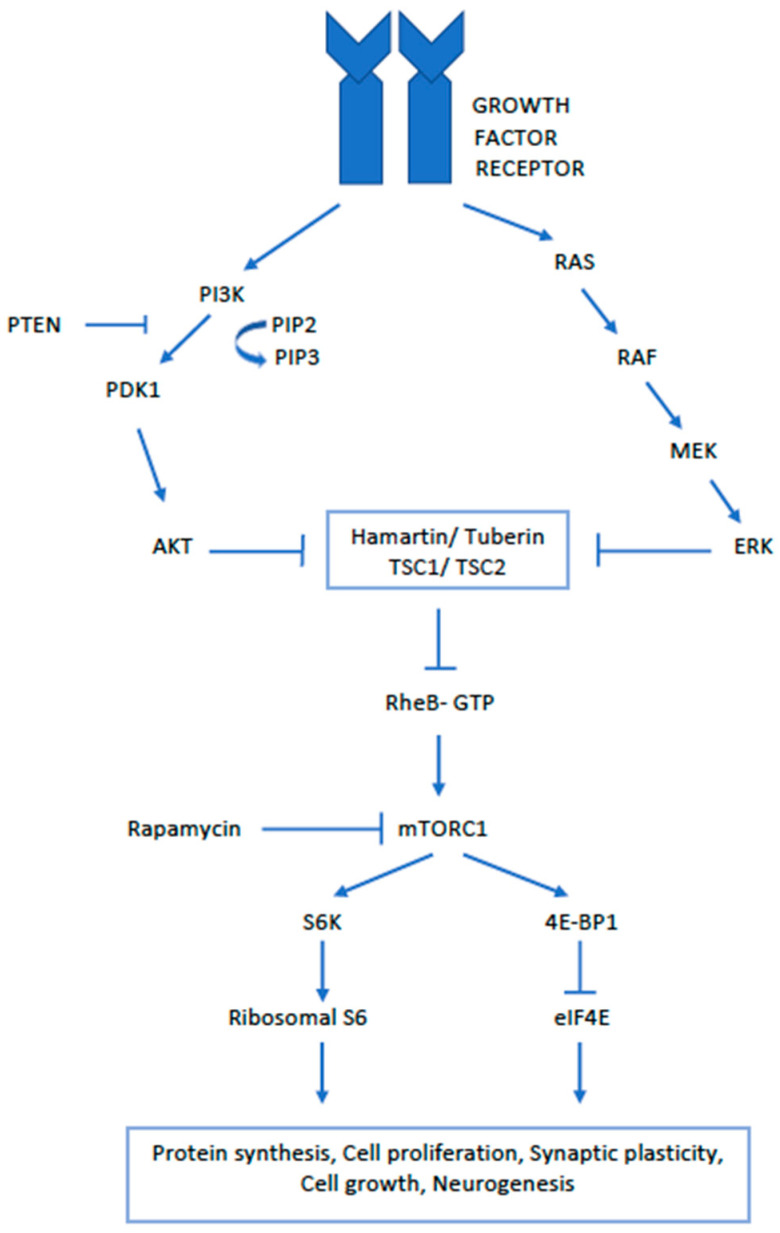
Schematic presentation of mammalian target of rapamycin (mTOR)-signaling pathway. The mTORC1 is activated by extracellular growth factors via two pathways: PI3K/AKT and Ras/RAF/MEK/ERK. The *TSC1/TSC2* is a critical integrator for upstream signals and functions as negative regulator of mTORC1. Rapamycin is an alternative negative regulator of mTORC1. The downstream signaling consists of activation of ribosomal biogenesis and protein synthesis through S6K and eIF4E.

**Figure 4 jcm-12-00956-f004:**
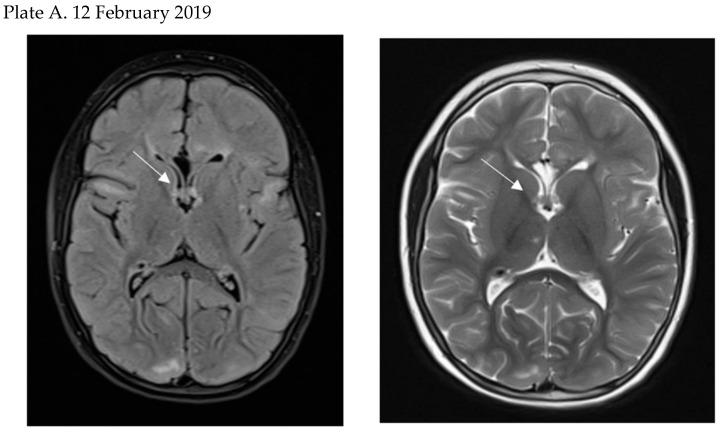
Growth of the SEGA in the proximity of foramen Monro during one year of follow-up (**plates A**, **B**, **C**). Pronounced hydrocephalus caused by growing SEGA is seen on plate C. This MRI image shows also a subcortical nodule in the right occipital lobe. (Courtesy of prof. Elzbieta Jurkiewicz, The Children’s Memorial Health Institute, Warsaw).

**Figure 5 jcm-12-00956-f005:**
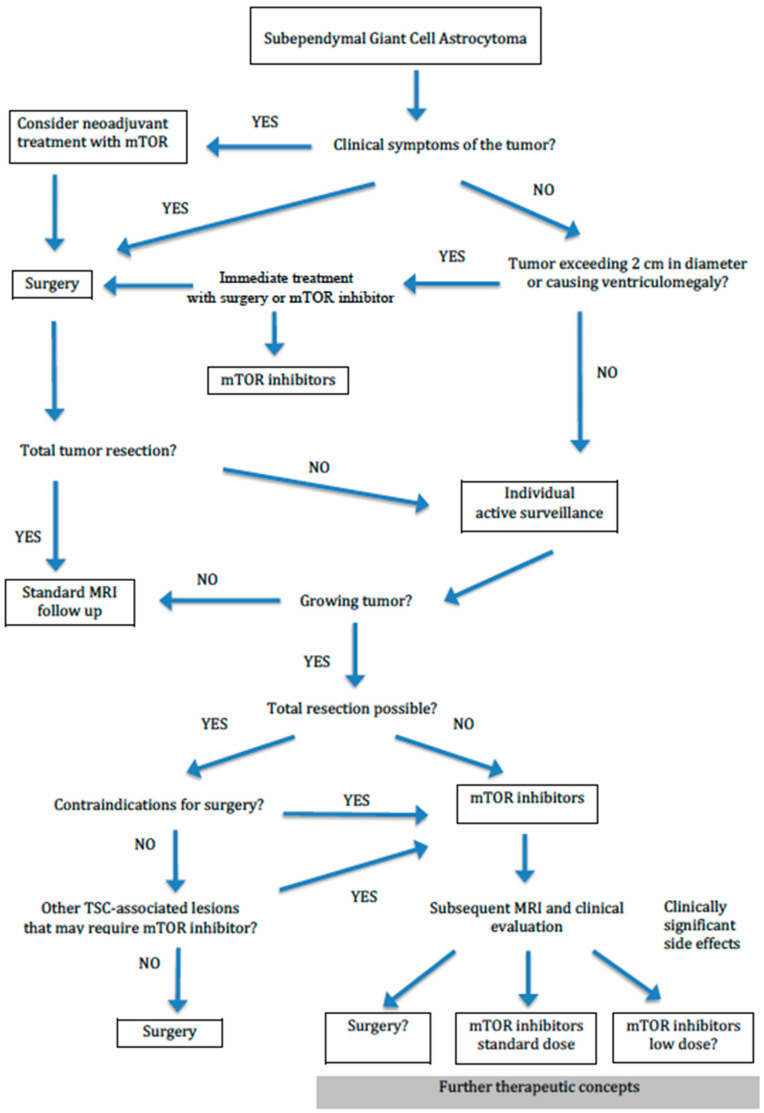
Algorithm of SEGA management (modified from Jozwiak et al. [7]).

**Table 1 jcm-12-00956-t001:** Most common side effects of surgical and medical treatments of SEGA.

Side Effects of Surgical Treatment	Side Effects of mTOR Inhibitors Treatment
Hemiparesis Hydrocephalus Memory deficits Intracranial bleeding Meningitis Subdural hygroma Basal edema	Stomatitis Pyrexia Diarrhea Nasopharyngitis Upper respiratory tract infection Vomiting Hypercholesterolemia Amenorrhoea Pneumonitis

**Table 2 jcm-12-00956-t002:** Indications for surgical and medical treatment of SEGA.

Indications for Surgery	Indications for mTOR Inhibitors
Acute symptomatic SEGA Asymptomatic SEGA with dilatation of ventricles Residual tumor after neoadjuvant treatment with mTOR inhibitor	4. Contraindications for surgery (e.g., very high protein in CSF, contraindication for general anesthesia, existing rhythm abnormalities), 5. Bilateral SEGAs 6. Difficulties with total resection (risk of regrowth) 7. Neoadjuvant therapy before surgical resection 8. Need for treatment of other manifestations of TSC

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
