# Peer review of "Subependymal Giant Cell Astrocytomas in Tuberous Sclerosis Complex—Current Views on Their Pathogenesis and Management"

_jcm, 2023, doi:10.3390/jcm12030956_

Round 1
Reviewer 1 Report
TSC has a variety of age-dependent complications. This review focuses on SEGA, one of the major manifestations of TSC, and is expected to attract the attention of many researchers, not only in the EU and North America, but also in Asian countries such as Korea, Japan, and China, where mTOR inhibitors are currently approved. The paper begins its description with the concept of TSC, and then describes the epidemiology of SEGA, as well as the medical treatment of SEGA with mTOR and the neurosurgical treatment of SEGA from a broad perspective. Overall, this review article is appropriately structured and does not require major revisions. A few minor comments are noted below, and the authors are encouraged to revise and respond.
There is a description of TSC1 and TSC2 symptoms in the Introduction such as in Ref. 4. Please consider a more detailed description, as basic animal studies have also reported that TSC2 is characterized by more severe cases of epilepsy. Tsc2 gene inactivation causes a more severe epilepsy phenotype than Tsc1 inactivation in a mouse model of tuberous sclerosis complex. Hum Mol Genet 20:445-454, 2011.
please consider an additional mention of auditory nerve tumors as a co-morbidity of TSC2.
In addition, there is an interesting literature on the increasing use of mTOR for epilepsy with TSC comorbidities. Please check it out. Improvement in Impaired Social Cognition but Not Seizures by Everolimus in a Child with Tuberous Sclerosis-Associated Autism through Increased Serum Antioxidant Proteins and Oxidant/Antioxidant Status. Case Rep Pediatr. 2019 Nov 23;2019:2070619.
There is no explanation regarding autism spectrum disorders, where the effects of mTOR have been noted. Please cite the literature and consider adding it. Contribution of Transferrin and Ceruloplasmin Neurotransmission and Oxidant/Antioxidant Status to the Effects of Everolimus: A Case Series.Cureus. 2020 Feb 8;12(2):e6920.
In introduction, please also explain the adjacent gene syndrome, PKD/TSC.
Could you create and add a schematic diagram of the mTOR signaling pathway? There are two pathways: one that starts with NMDA, TrkB, and the receptor, followed by PI3K, PTEN, PDK1, and AKT, and another that starts with IGF-1, followed by RAS, NF1, RAF, MEK, and ERK. Beyond that, there is TSC1/TSC2, the pathway through which mTOR acts.
What is the technique used to remove SEGA in neurosurgery? There is a newer, less invasive technique in which a pinhole is made in the skull and the SEGA is removed in a keyhole fashion.
In addition to the SEGA, the MRI image shows a subcortical nodule in the right occipital region.
The importance of mTOR as a neoadjuvant therapy for microscopic residual tumors will increase in the future. This point could be further emphasized.
Best regards,
Dr. Reviewer
Reviewer 2 Report
This article by Cho et al provides a review on the epidemiology, pathology, pathogenesis, diagnosis and treatment of subependymal giant cell astrocytoma (SEGA) associated with tuberous sclerosis complex. Overall. this review is succinct, clinically relevant, and nicely written. The treatment protocol (Figure 4) is appropriate and practical.
Prior to publication, several minor issues should be addressed:
1. About SEGA in adulthood, rare adult cases of not only sustained growth, but also of new occurrence of SEGA were reported by reference #39. The latter should be described.
2. There are several inadequate terms or phrases, such as “pathogenetic genetic findings” (line 68-69), “his growth” (line 138), “like for example in the third ventricle” (line 151), and “due their different clinical profile” (line 309), as well as too wide space between “of” and “mTOR” (Table 1) and unnecessary change of line (line 287-288).
